# Rational Design for the Complete Synthesis of Stevioside in *Saccharomyces cerevisiae*

**DOI:** 10.3390/microorganisms12061125

**Published:** 2024-05-31

**Authors:** Wei Huang, Yongheng Liu, Xiaomei Ma, Cilang Ma, Yuting Jiang, Jianyu Su

**Affiliations:** School of Life Science, Ning Xia University, Yinchuan 750000, China; xinzi6023@sina.com (W.H.); 13519512834@139.com (Y.L.); 18209607127@139.com (X.M.); macilang@163.com (C.M.); jyt_0831@163.com (Y.J.)

**Keywords:** *Saccharomyces cerevisiae*, stevioside, steviol, diterpenoid, terpenoid synthases, glycosyltransferases

## Abstract

Stevioside is a secondary metabolite of diterpenoid glycoside production in plants. It has been used as a natural sweetener in various foods because of its high sweetness and low-calorie content. In this study, we constructed a *Saccharomyces cerevisiae* strain for the complete synthesis of stevioside using a metabolic engineering strategy. Firstly, the synthesis pathway of steviol was modularly constructed in *S. cerevisiae* BY4742, and the precursor pathway was strengthened. The yield of steviol was used as an indicator to investigate the expression effect of different sources of diterpene synthases under different combinations, and the strains with further improved steviol yield were screened. Secondly, glycosyltransferases were heterologously expressed in this strain to produce stevioside, the sequence of glycosyltransferase expression was optimized, and the uridine diphosphate-glucose (UDP-Glc) supply was enhanced. Finally, the results showed that the strain SST-302III-ST2 produced 164.89 mg/L of stevioside in a shake flask experiment, and the yield of stevioside reached 1104.49 mg/L in an experiment employing a 10 L bioreactor with batch feeding, which was the highest yield reported. We constructed strains with a high production of stevioside, thus laying the foundation for the production of other classes of steviol glycosides and holding good prospects for application and promotion.

## 1. Introduction

Terpenoids are a large class of secondary metabolic natural products in nature. They are used in many applications in various aspects of life because of their various activities. Steviol glycosides (SGs) are glycosidically derived secondary metabolites of kaurene-type tetracyclic diterpenoids, mainly from the leaves of *Stevia rebaudiana* [1,2]. SGs is a general term for glucoside compounds in *S. rebaudiana*, and more than 30 types of SGs have already been isolated and identified [3]. The biosynthesis of SGs begins with the isoprene (IPP) of C5, which is converted into dimethylpyrophosphate (DMAPP) via the isomerase IDI1, and then combines three IPPs through a three-step condensation reaction to form geranylgeranylpyrophosphate (GGPP). In *S. rebaudiana,* IPP is synthesized via the 2-C-methyl-d-erythritol-4-phosphate (MEP) pathway, while in *Saccharomyces cerevisiae*, IPP is synthesized via the mevalonate (MVP) pathway [4]. GGPP then undergoes two cyclizing reactions to form kaurene via diterpene synthase copyl-pyrophosphate synthase (CPS) and kaurene synthase (KS). Next, under the action of kaurene oxidase (KO) and kaurenoic acid hydroxylase (KAH) from the NADPH-dependent cytochrome P450 oxidase family, there is sequential oxidation at the C19 and C13 positions, leading to the formation of kaurenic acid and the steviol glycoside skeleton (Figure 1) [5,6,7]. Finally, the transfer of different numbers and types of glycosyl molecules via glycosyltransferases at the C13 hydroxyl group and C19 carboxyl group of the steviol skeleton results in the formation of different glycosidic products, which causes the sweetness and taste of various glycosides to vary greatly [8,9]. Studies have shown that the greater the number of glycosylated molecules and the longer the side chains, the more enhanced the taste of SGs [10,11]. The most popular and better-tasting SGs are rebaudioside A, rebaudioside D, and rebaudioside M, which contain four, five, and six glucosides, respectively. Stevioside is slightly bitter and contains three glucosides. In *S. rebaudiana*, only four glycosylated transferases are involved in the modification of glucosylation, while producing more than 20 glycosides [12]. Glycosylation is thus very complicated and produces many by-products.

With the increasing demand for low-sugar and low-calorie products, the traditional method of extracting SGs from plants not only pollutes the environment but also cannot meet the market demand because of the low yield [13]. The creation of artemisinic acid-producing yeast has paved the way for obtaining more natural active products of plant secondary metabolism, which are subject to seasonal and environmental influences [14]. *S. cerevisiae* is among the earliest microorganisms and is closely related to human life. It is a model organism with a clear genetic background and biosafety certification allowing it to be used as a cell factory in many natural products [15].

In recent years, research on the production of SGs in microorganisms has yielded certain results. In 2016, Jiangfeng Wang [16] constructed the first engineered strain for the de novo production of rebaudioside A in *Escherichia coli*, obtaining a yield of 10.03 mg/L. However, this yield was too low, and *E. coli*, as a prokaryotic expression system for the heterologous synthetics of natural products, has some disadvantages, such as its lack of various modification functions after enzyme translation. Yameng Xu [17] used a systematic engineering strategy to de novo synthesize rubusoside and rebaudioside in *S. cerevisiae*. Amounts of 1368.6 mg/L rubusoside and 132.7 mg/L rebaudioside were produced in a 15 L bioreactor. Although the production of rubusoside containing two glucosides increased, glycosylation was also more difficult to achieve with the increased number of linked glycosides in the side chain, which was followed by an additional increase in by-products. Other studies have focused on the targeted modification of glycosyltransferases to screen for more specific enzymes to reduce the accumulation of by-products and intermediates. Yaping Mao [18] constructed a whole-cell catalytic strain for rubusoside, first using *S. cerevisiae* as the chassis cell; then, the simulation and targeted modification of SrUGT74G1 showed that the S84A/E87A double-mutant enzyme of SrUGT74G1 converted 72% of steviol into rubusoside without the accumulation of intermediate S19G. Nevertheless, whole-cell catalysis requires the addition of substrates and cofactors, and the reaction conditions are strictly controlled. In addition, SGs have a similar main structure and a complex glycosylation network. This makes it difficult to modify several glycosyltransferases to catalyze the glycosylation of specific sites of the stevia skeleton.

In this study, we attempted to engineer an *S. cerevisiae* strain which has capable of producing stevioside containing three glucosyl groups in high yield. The biosynthetic pathway of steviol was established using a combinatorial modular operation (Figure 1) to enhance the supply of GGPP by overexpressing endogenous key genes, knocking down the branching pathway, and fusing expression pathway enzymes. Using steviol as an indicator, the diterpene synthases from different sources and different combinations were screened to obtain the strain with the highest yield of steviol. Then we expressed the module of glycosyltransferases and optimized the expression sequences of three glycosyltransferases by using the directionally retrofitted UGT74G1*_S84A/E87A_* and SrUGT91D2e_NO.5, and strengthened the supply of UDP-Glc. Ultimately, the engineered strain producing a high yield of stevioside was obtained.

## 2. Materials and Methods

### 2.1. Strains and Reagents

*Saccharomyces cerevisiae* BY4742 was used as the initial strain, and *Escherichia coli* Top10 was used for plasmid construction. All strains used and constructed in the study are shown in Appendix A. Restriction endonucleases were purchased from NEB (New England Biolabs, Ipswich, MA, USA), Phusion DNA polymerase and assembly reagent ClonExpress MultiS One-Step Cloning Kit were purchased from Vazyme Biotech (Nanjing, China), and the plasmid extraction and yeast genome extraction kits were purchased from TIANGEN Biotech (Beijing, China). *Saccharomyces cerevisiae* classic transformation kit was purchased from Coolaber Technology (Beijing, China). The standard of steviol and stevioside were purchased from Yuanye Biotechnology (Shanghai, China).

### 2.2. DNA Manipulation

The heterologous genes *SrCPS* (KT276239) from *Stevia rebaudiana*, *SrKS22*-1 (AF097311) from *Stevia rebaudiana*, *SrKO* (CYP701A5) from *Stevia rebaudiana*, *SrKAH* (EU722415) from *Stevia rebaudiana*, *SrCPR1* (ABB88839.2) from *Stevia rebaudiana*, *SrUGT85C2* (AY345978) from *Stevia rebaudiana*, *SrUGT91D2e* (AY345980) from *Stevia rebaudiana*, *SrUGT74G1* (AY345982) from *Stevia rebaudiana*, *AtCPS* (NM_116512) from *Arabidopsis thaliana*, *AtKS* (NM_106594) from *Arabidopsis thaliana*, *ZmtCPS* (AY562490) from *Zea mays*, *ZmKS* (NP_001105097) from *Zea mays*, *GfKS* (Q9UVY5.1) from *Gibberella fujikuroi*, *PpKS* (BAF61135) from *Physcomitrium patens*, and *AgKS* (AAB05407) from *Abies grandis*. After being code-optimized, they were synthesized via Sangon Biotech (Shanghai, China). Homologous recombinant plasmids pUSC-01 and pUSC-02 were used for the gene integration of heterologous gene modules. The plasmids contain two *LoxP* sequences in the same direction, and the middle of the *LoxP* sequence contains the replication initiation site, the resistance gene *Kan^R^*, and the yeast screening gene *URA3*. The two isotropic *LoxPs* can be recombined during no-selective-pressure subculturing, which results in the loss of the marker gene *URA3*. The plasmid can be recombined many times without a trace. The recombinant plasmids are shown in Appendix A, the plasmid construction method is referred to in [19], and the primers and constructed plasmids are shown in Appendix A. The promotors and terminators in the experiment were amplified from the genome of BY4742 via polymerase chain reaction (PCR); TEF1, TEF2, PGK1, TDH3, and HXT1 promoters; and CYC1, ADH1, TEF1, TEF2, and PGK1 terminators. Prior to plasmid assembly, small fragments were assembled using overlap extension PCR (OE-PCR, 15–20 bp overlaps of each adjacent fragment were used to implement homologous assembly), purified, and subsequently equimolar-mixed using a ClonExpress MultiS One-Step Cloning Kit (Vazyme Biotech, Nanjing, China). After the reaction at 37 °C for 30 min, then they were transformed into *E. coli* Top10. The chromosomal integration locus in this study was selected from a study by Reider Apel [20].

### 2.3. Strain Construction, Transformation, Screening, and Culture

Successfully constructed and correctly sequenced recombinant plasmids were linearized and purified; then, about 1 μg of DNA was transformed into *S. cerevisiae* cells using the LiAc/SS carrier DNA/PEG method (*S. cerevisiae* classic transformation kit). The transformed cells were centrifuged, collected, suspended in sterile water, coated on a synthetic dropout URA medium (SD-URA) screening plate, and cultured at 30 °C for 4 d. Single colonies were selected from the plate for colony PCR verification with detection primers (the detected primers are listed in Appendix A). The positive strains were repeatedly coated on the SD-URA screening plate for two generations. The screened positive strains were inoculated in the liquid yeast extract peptone dextrose (YPD) medium for 24 h and then coated on the synthetic dropout medium with 5-fluoroorotic acid (SD-FOA) counterselection plate. Finally, the positive strain that was successfully recombined grew on the plate.

*E. coli* strains were cultured at 37 °C in Luria-Bertani (LB) medium (0.5% yeast extract, 1% tryptone, and 1% NaCl), supplemented with 50 μg/mL of kanamycin or 100 μg/mL of ampicillin. YPD medium (1% yeast extract, 2% tryptone, and 2% glucose; solid plate: 2% agar powder) was used to cultivate *S. cerevisiae.* SD-URA was used to engineer the yeast strain (purchased from Coolaber Technology, Beijing, China). A package of solid powder was taken and dissolved in 500 mL of ultra-pure water. A synthetic complete medium containing 1 mg/mL of 5-fluoroorotic acid (5-FOA; 1 mg of 5-FOA was dissolved in 1 mL of DMSO and filtered using a 0.22 μm sterile filter membrane) was used to counterselect the engineered yeast strains with URA3 marker excision.

### 2.4. Metabolite Extraction and Quantification

The target compounds were characterized and quantified using liquid chromatography–tandem mass spectrometry (LC-MS) and high-performance liquid chromatography (HPLC) with the external standard method, respectively. A total of 5 mL of the 96 h fermentation broth was taken, centrifuged at 8000 rpm for 5 min, and the supernatant was collected. Anhydrous ethanol was added to the supernatant to a punch concentration of 70%, and the extract was ultrasonicated at 60 °C in a water bath for 30 min and subsequently centrifuged at 2400 rpm for 10 min. The extraction was repeated twice, and the combined supernatants were concentrated to 5 mL via rotary evaporation, which was the sample of the extracellular product, and then filtered through a 0.22 μm filter membrane for HPLC detection. The cell precipitate was washed twice with sterile water, and the cells were resuspended by adding 5 mL of sterile water and frozen at −80 °C overnight, followed by the addition of 20 mg of snail enzyme and thorough mixing; then, a wall-breaking treatment was carried out in a warm bath at 30 °C for 1 h. At the end of the process, the supernatant was collected via centrifugation, and the same extracellular sample was treated with ethanol extraction and concentrated. Production was then detected.

LC-MS was conducted using a ThermoFisher LTQ Orbitrap XL (Thermo Scientific, Wilmington, NC, USA) equipped with a Accucore C18 column (2.1 × 150 mm, 2.6 μm) (Thermo Scientific, Wilmington, NC, USA). Samples were eluted at 35 °C using the following gradient program with solvent A (acetonitrile) and solvent B (0.1% formic acid) as the mobile phase at a flow rate of 0.2 mL/min: 0–4 min, 80–50% B; 4–7 min, 50–0% B; 7–8 min, 0% B hold; 8.01 min, 80% B. MS was set to electrospray ionization (ESI) in negative mode. Full MS was set as follows: 100–1500 *m*/*z*. For quantitative analysis of production using HPLC, a WATERS 2695 (Waters Corp., Milford, MA, USA) with diode array detector (DAD) was utilized with the check wavelength set to 210 nm and an InertSustain NH_2_ column (4.6 × 250 mm, 5 μm, flow rate: 1 mL/min, column temperature: 35 °C) (GL Sciences, Tokyo, Japan). Samples were eluted using the following gradient program with solvent A (acetonitrile) and solvent B (water): 0–15 min, 95–82% B; 15–20 min, 82% B hold; 20.01 min, 95% B; 20–27 min, 95% B hold.

### 2.5. Fed-Batch Fermentation of Yeast Strain SST-302III-ST2

Strain activation: the glycerol-preserved engineered strain SST-302III-ST2 was seeded onto a YPD plate, cultured at 30 °C for 3 d, inoculated into 100 mL flasks containing 10 mL of liquid YPD at 30 °C at 200 rpm, and then cultured for 20 h. Preparation of primary seed solution: a single colony of the activated strain was taken at 5% inoculum dosage and inoculated into 250 mL flasks containing 50 mL of liquid YPD and then cultured at 200 rpm at 30 °C for 20 h. Secondary seed solution preparation: the primary seed solution was taken with a 5% inoculum quantity, inoculated into 500 mL flasks containing 100 mL of liquid YPD, and then cultured at 30 °C at 200 rpm for 20 h. Fermenter inoculation: The secondary seed solution (250 mL) was taken with an initial OD_600_ of about 0.5 and a 10% inoculum quantity, and inoculated into a 500 mL triangular flask containing 100 mL of liquid YPD. The secondary seed solution was inoculated into a 10 L fermenter (Huisen Biological Equipment, Zhejiang, China) containing 6 L of liquid YPD for fermentation. The fermentation temperature was 30 °C, and a pH of 5.5 was maintained with NH_4_OH automatically added. The ventilation rate is 2vvm, and the stirring speed is adjusted by correlating the stirring speed and dissolved oxygen (DO), the agitation speed was maintained at 200–500 rpm, and DO is controlled above 35% by stirring speed. The total fermentation time was 168 h.

Fermentation was carried out for 20 h for feeding purposes. The supplement ingredients included the following [18,21]: 500 g/L of glucose, 9 g/L of KH_2_PO_4_, 2.5 g/L of MgSO_4_, 3.5 g/L of K_2_SO_4_, 0.28 g/L of Na_2_SO_4_, 10 mL/L of a trace element solution, and 12 mL/L of a vitamin solution. After the initial feeding, a yeast powder with a final concentration of 8 g/L was added every 10 h as a nitrogen source.

## 3. Results

### 3.1. Construction of S. cerevisiae to Produce Steviol

Because of the biosynthetic pathway of the precursor compound, GGPP exists in *S. cerevisiae*. The genes *SrCPS*, *SrKS*, *SrKO*, *SrKAH*, and NADPH-dependent cytochrome P450 reductase *SrCPR* of the SGs biosynthesis pathway, derived from *S. rebaudiana* via codon optimization and synthesis. For the cyclization module *SrCPS* and *SrKS*, together with their promoters and terminators, were constructed as the recombinant plasmid p1309CK(pUt301I), which was integrated at the *ARS1309a* site in the genome of *S. cerevisiae* BY4742. For the oxidation module, *SrKO*, *SrKAH,* and *SrCPR* were combined to form the recombinant plasmid pYPL, which was inserted into the *YPL062w* site, because previous studies have shown that the production of Acetyl-CoA and terpenoids can be increased after YPL062w elimination [22]. After these two rounds of integrations, strain SST-001 was formed. After 96 h of culturing strain SST-001, the steviol signal [M-H]^−^ = 317.212 was detected via LC-MS; however, no signal of steviol was detected via HPLC (Figure 2A). This may be because the amount of strain metabolic flow to GGPP was only used for strain growth, and there was not enough GGPP to become the substrate for SrCPS, which is consistent with the results reported in previous studies [23].

Thus, the synthesis pathway of the precursor compound GGPP was strengthened to ensure the strain produced steviol. Combined with reports from the existing research, we made the following modifications to the precursor pathway of strain SST-001 (Figure 2B): one copy of the truncated hydroxymethylglutarate monoyl CoA reductase (*tHMG1*) was overexpressed, and the *ERG9* promoter in the branching pathway was replaced with a glucose concentration-responsive promoter, P*_HXT1_*, which downregulated the expression of downstream genes as glucose concentration decreased [24]. The combined module described above integrated at the promoter position of *ERG9*, formed strain SST-101. LC-MS was used to detect the characteristic signal of stevioside in the strain, and the steviol detected in the shake flask culture was 25.97 mg/L (Figure 2C,D). Then we overexpressed the second *tHMG1* gene and the mutant *UCP2-1* of the transcriptional regulatory factor *UCP2*, integrated the module at the *YJL064w* site in the chromosome of strain SST-101, and formed strain SST-201. Detected 126.1 mg/L of steviol in the shake flask culture (Figure 2C,D), 4.86 times higher than SST-101.At last, we overexpressed isomerase IDI1 (isopentenyl diphosphate: dimethylallyl diphosphate isomerase, IDI1), which was converted from IPP to DMAPP, and carried out the fusion expression of *ERG20* and its mutant, *mERG20^F96C^*. The module was integrated at the *YPLdelta15* site in the chromosome of strain SST-201, formed strain SST-301. We detected 166 mg/L of steviol, representing a yield 1.32 times higher than SST-201, and 6.39 times higher than SST-101 (Figure 2C,D). Most of the steviol was detected in the fermentation broth supernatant.

The results show that over-expressing key genes, fusion expression, and downregulating the expression of branching pathway genes effectively improve the metabolic flux of the MVA pathway. These operations also greatly increased the steviol content, laying the foundation for the subsequent heterologous synthesis of stevioside.

### 3.2. Studies on Diterpenoid Synthases

The catalytic efficiency of enzymes is a key factor in the high yield of target compounds, and the screening of enzymes from different sources is also an effective strategy to increase the yield of specific host heterologous pathways [25,26]. Different diterpene compounds are formed from GGPP and catalyzed via various diterpene synthases. Both steviol and gibberellinic acid are ent-kaurene-type diterpene skeletons [27]. In *S. rebaudiana*, they are both formed from GGPP via the two-step cyclization of -(-)ent-copalyl diphosphate synthase (CPS) and -(-)ent-kaurene synthase (KS). CPS and KS are members of diterpene synthase (diTPS) in the terpene synthase family [28]. CPS is a class II diterpene synthase, and KS is a class I diterpene synthase. Their primary structure is similar, but the CPS structure contains a βγ domain, with a conserved DXDD motif at the active site as a proton donor to GGPP. KS contains an α domain, with a DDXXD motif in its active site that is used for Mg^2+^ ionization and binding to ent-CPP (Figure 3A). There is also a class of bifunctional terpenoid synthases in nature, which contain an αβγ domain and can catalyze two steps of cyclization continuously, such as Grand Fir (*Abies grandis*) of AgAS and *Physcomitrella patens* of PpKS [29].

We used the steviol yield as an indicator, selected diterpene synthases from different sources, and tested the effects of different combination patterns on expression to further improve steviol yield. This is because the fusion expression of the two enzymes catalyzing the sequential reaction using flexible linking peptides can improve the binding efficiency of the substrate to the enzyme and minimize the loss of intermediate metabolites [30,31]. In this study, class I/II diterpene synthases, such as SrCPS, SrKS, AtCPS, AtKS, ZmtCPS, and ZmKS, and bifunctional diterpene synthases, such as GfKS, PpKS, and AgKS, were selected. The amino acid sequences of the selected diterpene synthases were compared with SrCPS and SrKS via DNAMAN (Appendix A). The results showed that all the selected diterpene synthases had active catalytic functional regions. The selected genes formed a combination of modules with different patterns (Figure 3A). Combinations II and III involved fusion expression, with the CDS region of the two genes linked via a flexible binding peptide, “GGGS”. There was a sequence exchange between the genes of combinations III and II. Expression module IV represents the bifunctional diterpene synthases, 310 IV, 311 IV, and 312 IV represent GfKS, PpKS, and AgKS, respectively.

The yields of all of the strains are shown in Figure 3B. No steviol was produced in SST-312 IV. It has been reported that AgAS mainly catalyzes GGPP-formed nor-CPP, and then cyclizes it to form a rosin-type diterpene skeleton [32]. Other diterpene synthases and their combinations produced steviol. Strain SST-302 III had the highest yield, which involved the fusion expression of the class I diterpene synthase AtKS from *Arabidopsis thaliana* and the class II diterpene synthase SrCPS from *S. rebaudiana*, yielding 217.2 mg/L of steviol and being 30.84% higher than SST-301 (Figure 3B). In combinations II and III, the fusion protein brings the active centers of the two sequential enzymes closer together. In combination III, after exchanging the fused sequences of class I diterpene synthases and class II diterpene synthases, the activity centers of class I enzymes were closer to those of class II enzymes (Figure 3A). The strain of combination III had the highest yield of steviol compared to all combinations (Figure 3B), followed by combination II, with combination I having the lowest yield. Steviol production in the combined strains SST-307, SST-308, and SST-309 containing the class II diterpene synthase ZmtCPS was lower than that of the combined strains SrCPS and AtCPS; the catalytic efficiency of SrCPS and AtCPS was comparable. Steviol production in SST-303 and SST-306 strains with the class I diterpenoid synthase ZmKS was lower than that of SrCKS and AtKS; the catalytic efficiency of SrKS and AtKS was comparable. In the bifunctional diterpene synthases, GfKS and PpKS both produced steviol, yielding 159.46 mg/L and 75.45 mg/L (Figure 3B), respectively. These results indicate that both of these bifunctional diterpene synthases can form kaurene diterpenes. SST-302 III, which had the highest yield of steviol, was used for follow-up experiments.

### 3.3. Complete Biosynthesis of Stevioside in S. cerevisiae

To obtain a strain-producing stevioside, we integrated the glycosyltransferase modules (*SrUGT85C2*, *SrUGT91D2*, and *SrUGT74G1*) into the genome of strain SST-302 III (*ARS1414a* site), thereby forming the strain SST-302III-ST. Following shake flask culture for 96 h, we detected the stevioside signal [M-H]^−^ = 803.369 via LC-MS in the fermentation broth of SST-302III-ST, which indicates that the glycosyltransferase module was successfully expressed (Figure 4A). However, only 17.43 mg/L of stevioside was detected in the fermentation broth, with 137.5 mg/L of steviol also detected in the fermentation broth (Figure 5B). Full-scan MS also detected signals [M-H]^−^ = 479.264 and [M-H]^−^ = 641.316, which correspond to steviolmonoside (steviol-19-O-glucoside) and steviolbiside (rubusoside), respectively (Figure 4A). This means that steviol was not completely glycosylated, and there was a large accumulation of the intermediate products steviolmonoside (steviol-19-O-glucoside) and steviolbiside (rubusoside). This result is far from what we expected.

To ensure correct plasmid assembly and gene expression, we extracted the total RNA of the engineered strain SST-302III-ST, reverse transcribed the mRNA into cDNA, and amplified the gene fragments according to the CDS sequence of the introduced genes *SrCPS*, *SrKS*, *SrKO*, *SrKAH*, *SrCPR*, *SrUGT85C2*, *SrUGT91D2,* and *SrUGT74G1*. In addition, the total RNA of the original strain BY4742 was extracted, and reverse transcription amplification was used as a negative control. As can be seen from Figure 4B, all eight of the introduced heterologous genes were expressed successfully. The primer Premier 5 was used to design more specific primers, so the amplified products are all shorter than the original gene length.

It has been reported that complex networks of glycosylation are involved in the synthesis pathway of steviol glycosides (Figure 4C). In *Stevia rebaudiana*, only four glycosyltransferases catalyze all glucosylation reactions, and there are more than twenty glucosylation products [32,33]. In order to obtain strains that yield high amounts of stevioside, we decided to adjust the glycosylation sequence, replacing the original *SrUGT74G1* with *SrUGT74G1_S84A/E87A_*, which was directionally modified [18]; to reduce the accumulation of intermediate by-products, *SrUGT85C2* and *SrUGT74G1_S84A/E87A_* were then fusion expressed using the flexible peptide “GGGS”, and *SrUGT91D2e_NO.5* was replaced with *SrUGT91D2* [16]. This process formed the strain SST-302III-ST1, which yielded 55.6 mg/L of stevioside, but 90 mg/L of steviol remained in the fermentation solution (Figure 5A,B). Not all steviol was converted, and neither steviolmonoside (steviol-19-O-glucoside) nor steviolbiside (rubusoside) intermediates were converted to stevioside. This result is most likely due to the insufficient intracellular UDP-glucose donors [34,35].

In yeast cells, incoming glucose is phosphorylated to glucose-6-phosphate via hexokinase (HXK1), then converted to glucose-1-phosphate via phosphoglucomutase (PGM2), and, finally, condensed with uracil triphosphate-glucose (UTP) to form UDP-Glc via UDP-glucose pyrophosphorylase (UGP1). In order to enhance the intracellular UDP-glucose donor supply and improve substrate conversion, we consulted the relevant literature [17,36], and modularized the overexpression of *HXK1, PGM2*, and *UGP1* in the strain SST-302III-ST1 under the regulation of endogenous strong promoters PGK1, TDH3, and TEF2, thereby forming the strain SST-302III-ST2. The final yield of stevioside was 164.89 mg/L after 96 h of shaker fermentation, and the yield increased by 2.96 times compared with that of SST-302III-ST1 (Figure 5A,B).

To explore the influence of the expression of many heterologous genes on cell growth, we used the starting strain BY4742 as a control to detect the strain density during the fermentation process of the engineered strains SST-001, SST-301, SST-302III-ST, and SST-302III-ST2, and then created a growth curve (Figure 5C). The curve shows that, compared with BY4742, the lag time of the engineered strains is slightly longer. The biomass of SST-001 at the logarithmic and stable stages was the same as that of the starting strain, and even slightly higher at the end of the fermentation period. However, the biomass of the engineered strains SST-301, SST-302III-ST, and SST-302III-ST2 decreased compared with BY4742 at the stable stage.

This may be due to the large number of constitutive promoters used during the experiments, which would have caused an imbalance between cell growth and product accumulation. When constructing the gene expression module, the endogenous constitutive promoter of *Saccharomyces cerevisiae* was selected because the constitutive promoter can stably express heterologous genes [37]. When involving the synthesis process of complex compounds, it is necessary to express multiple heterologous genes in chassis cells. This results in too many constitutive promoters being used and too many heterologous genes being expressed in large quantities during the early stage of cell growth. These genes will compete to consume the energy generated by the cells, break the metabolic balance of the cells themselves, and eventually lead to increased cell lag and decreased biomass, resulting in the final output of the target products being affected [38].

### 3.4. Batch Feeding Culture in a 10 L Bioreactor

To produce larger amounts of stevioside, the strain SST-302III-ST2 was batch-fed fermented in a 10 L bioreactor, and we plotted the growth curve, the yield accumulation curve of steviol and stevioside, and the consumption curve of glucose (Figure 6). At the beginning of fermentation, 20 g/L of glucose was added as a carbon source to make the strain grow rapidly. The glucose was exhausted at 20 h; then, the flow filling of glucose was started, and the concentration of glucose in the fermentation broth was always lower than 5 g/L. The strain entered a stable period at 60 h, until the end of fermentation, with OD_600_ up to 91.96. At 20 h, we detected 5.27 mg/L of steviol. This yield gradually increased, the accumulation of steviol reached its maximum at 233.64 mg/L at 90 h, and the amount gradually decreased until the end of fermentation, with 5.98 mg/L remaining. We detected 2.55 mg/L of stevioside at 40 h. The yield of stevioside increased exponentially after 80 h, reaching 1104.49 mg/L by the end of fermentation at 168 h. This is one of the higher yields reported in the study of the heterologous production of stevioside from *S. cerevisiae*, compared with those obtained for rubusoside and rebaudioside A, which contain two and four glucose groups, respectively. This is because there are no reports on its yield as an intermediate product. Although stevioside is not a popular SG, its high yields also form the base for the production of other steviosides.

## 4. Discussion

In this study, a biosynthetic pathway for steviol was established, and an engineered strain with high steviol production was obtained by enhancing the precursor pathway. In order to further obtain high-yielding strains, classes I and II diterpenoid synthases from *S. rebaudiana*, *A. thaliana,* and *Zea mays*, as well as the yield of steviol under different combinations, were investigated. We found that in combinations II and III, which expressed the two sequentially reacted enzymes, the yield of steviol was higher than that expressed by two enzymes expressed alone; in combination III, the active centers of the two enzyme exchange sequences were closer than that in combination II. Some researchers have found that when SmCPS and SmKSL are fused and analyzed, the active site is closer, and the corresponding product yield is higher [26], which is consistent with our research. Due to the combined expression strain of combination I, after the two enzymes were expressed under the control of their respective promoters, they dispersedly existed in different locations within the cell, resulting in the incomplete catalysis of the intermediate product.

Glycosylation is a natural process as well as a key modification that takes place during biosynthesis and is one of the most common modifications [39]. Glycosylation enhances the solubility and stability of natural products, promotes their storage and accumulation in plant cells, and is also one of the main factors determining the biological activity and bioavailability of natural products [40]. Uridine diphosphate (UDP) glycosyltransferase is a large family of enzymes [41]. In *stevia rebaudiana*, there are several glycosylation reactions that take place during the late stages of stevioside synthesis.

In the strain SST-302III-ST, which was formed by the introduction of the glycosyltransferase module, the yield of steviol was only 17.43 mg/L. We verified that the coding gene of the introduced glycosyltransferase was correctly expressed by RT-PCR. Our expected synthesis pathway from steviol to stevioside is shown with the red arrow in Figure 4C. The glycosylation sequence of steviol is as follows: first, two glucose groups in the sequence were connected at the C13-OH position; then, one glucose group was connected at the C19-COOH position to form stevioside. However, the stevioside yield was too low, and the LC-MS results showed the accumulation of intermediate products, such as steviol, steviolmonoside (steviol-19-O-glucoside), and steviolbiside (rubusoside) in the fermentation broth. These results confirm the poor specificity of glycosyltransferase substrates involved in the biosynthesis of stevioside [5]. Specifically, in cells, UGT85C2 glucosylates the steviol backbone at the C13-hydroxyl position, forming a β-d-glucoside, whereas UGT74G1 glucosylates the C19-carboxylic acid functional group, giving rise to the formation of an ester, UGT91D2, which catalyzes the glucosylation of the glucose moieties directly bound to the C13 position via the formation of 1,2-β-d-dglucosidic linkages. UGT76G1, which was not involved in this study, catalyzes the second glucose group at the C19 position [12].

In order to reduce the accumulation of intermediate products and obtain a high yield of stevioside, we changed the glycosylation sequence. Using directionally modified SrUGT74G1*_S84A/E87A_* and SrUGT91D2e_NO.5, along with the fusion expression of SrUGT74G1*_S84A/E87A_* and SrUGT85C2, we reduced the accumulation of steviolmonoside (steviol-19-glucoside), with steviolbiside (rubusoside) forming first. Subsequently, stevioside was catalyzed via SrUGT91D2e_NO.5. Ultimately, the yield of stevioside reached 164.89 mg/L after enhancing the endogenous UDP-Glc supply.

In this study, although we engineered a strain of *S. cerevisiae* with a high stevioside yield, the taste is inferior to that of rebaudioside A (four glucose groups), rebaudioside D (five glucose groups), and rebaudioside M (six glucose groups). Studies have shown that the greater the number of side-chain glucose groups, the better the taste. However, with the increase in glucosyltransferase, glycosylation becomes more complex, and more by-products are produced. In the future, we plan to achieve the targeted modification of glycosyltransferases to enhance the recognition ability of specific substrates, thus engineering strains with a single rebaudioside.

In addition, by comparing the growth curves of a series of engineered strains, we found that the constitutive promoter used in this study placed a metabolic burden on strains [38]. This is due to the fact that multiple heterogeneous genes are expressed in the early stages of strain growth when the constitutive promoters are regulated, thus resulting in excessive energy which affects the accumulation of strain biomass. Furthermore, it also affects stevioside accumulation. Therefore, in future work, we plan to design a switch that can strictly regulate the expression of heterologous genes, to separate the cell growth stage from the product accumulation stage [42].

In conclusion, we used a series of synthetic biological strategies to heterogeneously express stevioside biosynthetic pathways in *S. cerevisiae* and obtain a high-yielding strain. This study provides insights into the production of other natural products and establishes the foundation for the production of other SGs using this engineered strain, which serves as a chassis cell for producing stevioside.

## Figures and Tables

**Figure 1 microorganisms-12-01125-f001:**
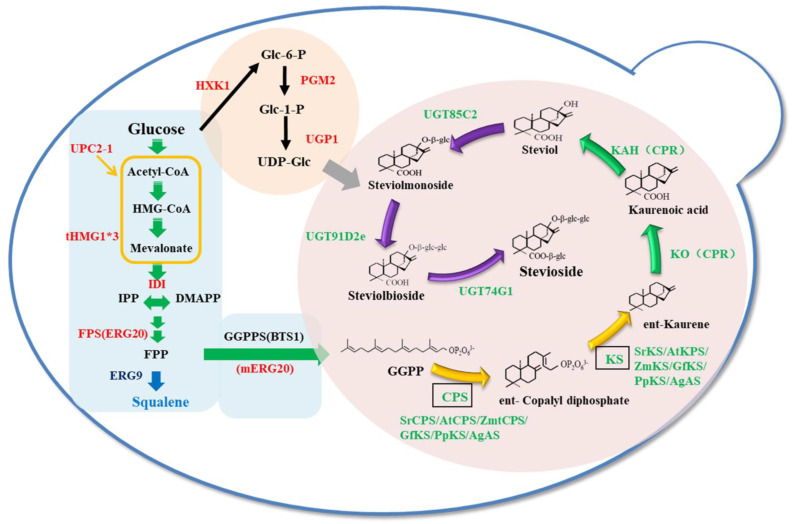
Schematic diagram of constructing a metabolic engineering strain with complete biosynthesis of stevioside in *Saccharomyces cerevisiae*. Green shows the introduction of heterologous genes; red shows the overexpression of endogenous genes, the tHMG1*3 represents that tHMG1 overexpressed 3 copies in this study; and blue shows the weakening of endogenous branching pathway genes. The CPS and KS framed in green were studied using different sources and different combinations. UCP2-1, transcriptional regulation factor; tHMG1, truncated hydroxymethylglutaryl-CoA reductase; IDI1, isopentenyl diphosphate:dimethylallyl diphosphate isomerase; FPS(ERG20), farnesyl pyrophosphate synthetase; GGPPS(BST1), geranylgeranyl diphosphate synthase; mERG20, F96 mutant of farnesyl pyrophosphate synthetase; IPP, isopentenyl diphosphate; DMAPP. dimethylallyl; GPP, geranyl diphosphate; FPP, farnesyl diphosphate; GGPP, geranylgeranyl pyrophosphate; ERG9, squalene synthetase; CPS, ent-copalyl diphosphate synthase; KS, ent-kaurene synthase; KO, ent-kaurene oxidase; KAH kaurenoic acid 13α-hydroxylase; CPR, NADPH-dependent cytochrome P450 reductases; UGT85C2, UDP-glycosyltransferase 85C2; UGT91D2e, the mutant of UDP-glycosyltransferase 91D2; UGT74G1, UDP-glycosyltransferase 74G1; HXK1, hexokinase isoenzyme 1; PGM2, phosphoglucomutase 2; UGP1, UDP-glucose pyrophosphorylase; Glc-6-P glucose-6-phosphate; Glc-1-P, glucose-1-phosphate; UDP-Glc, uridine diphosphate-glucose.

**Figure 2 microorganisms-12-01125-f002:**
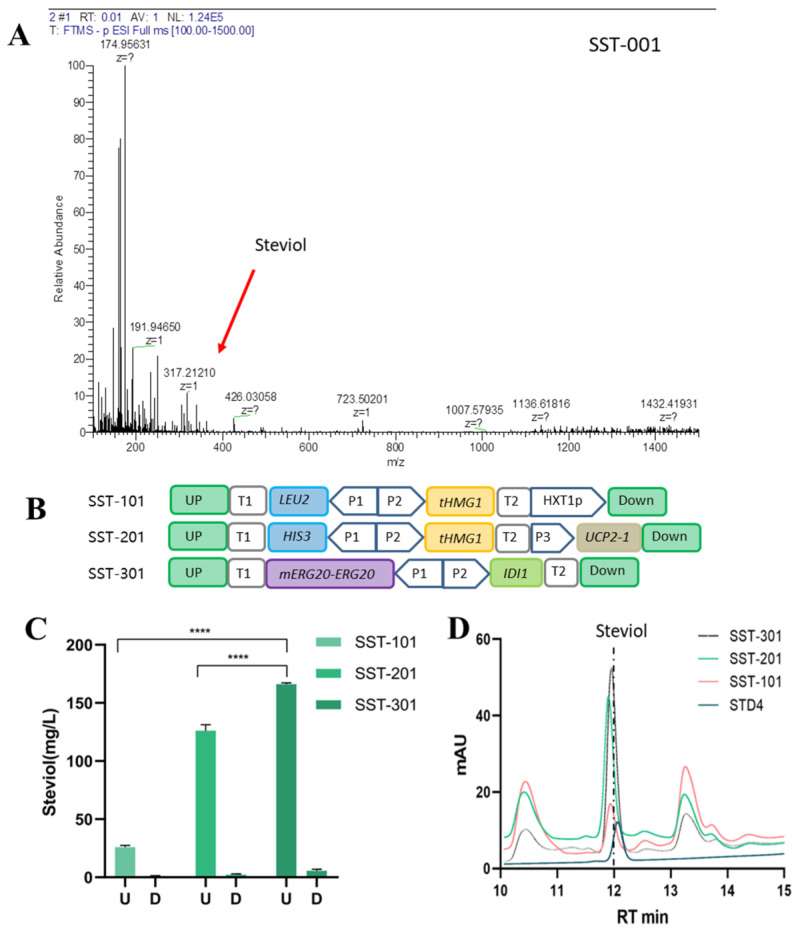
Steviol production in *S. cerevisiae*. (**A**) LC-MS diagram of fermentation broth of strain SST-001, the target signal is [M-H]^−^ = 317.212. (**B**) Schematic diagram of module combination during precursor pathway strengthen. (**C**) Steviol yield detected in fermentation broth of strains SST-101, SST-201, and SST-301 resulting from precursor pathway enhancement, U is the supernatant and D represents cell precipitation(*p* value < 0.0001 is labeled as ****). (**D**) HPLC of fermentation products of strains SST-101, SST-201, and SST-301, STD-4 is the standard of steviol.

**Figure 3 microorganisms-12-01125-f003:**
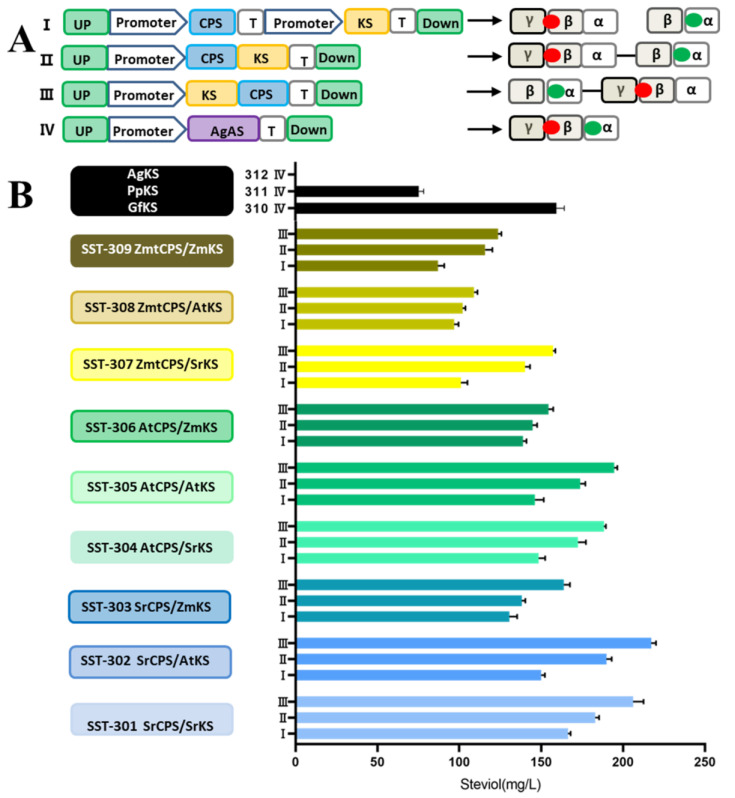
Studies of diterpenoid synthases. (**A**) Different combinations of diterpene synthases and the distance from active sites under different combination patterns. The class II diterpene synthase contains αβγ subunit, and the red circle in the middle of the βγ subunit is the active site; the class I diterpene synthase contains αβ subunit, and the green circle in α subunit is the active site. (**B**) Steviol yield of different diterpene synthases under different combination modes.

**Figure 4 microorganisms-12-01125-f004:**
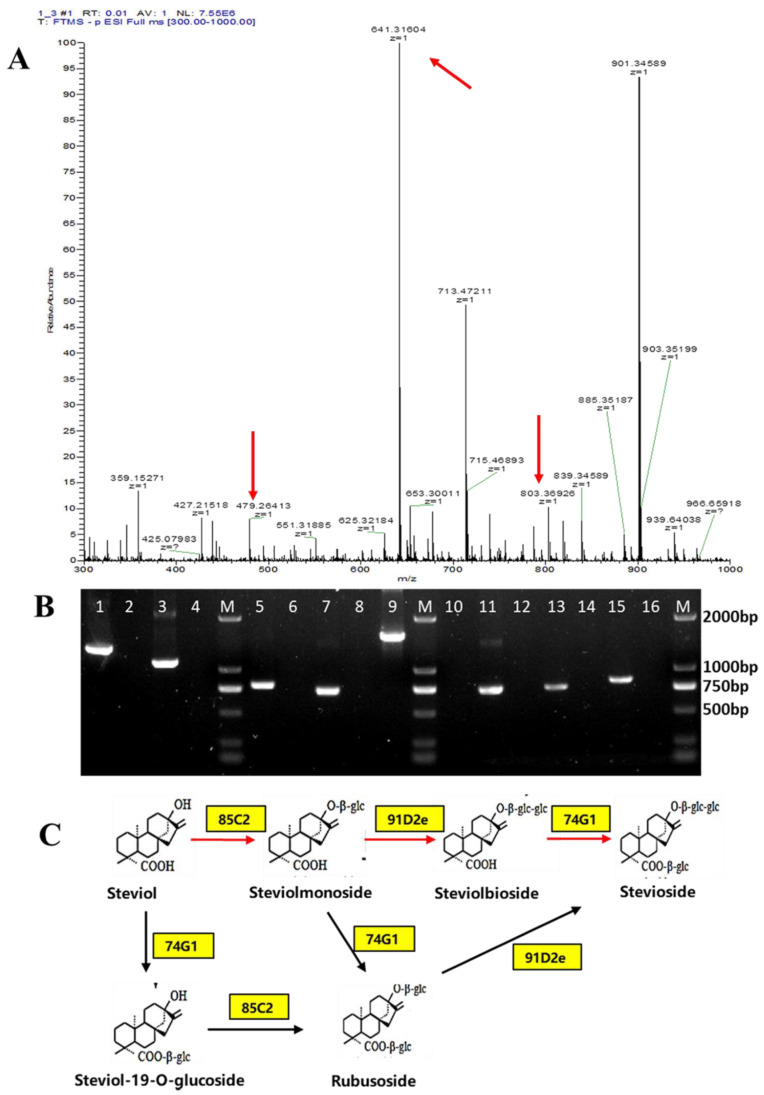
Complete biosynthesis of stevioside in *S. cerevisiae*. (**A**) LC-MS full-scan mode of fermentation broth of strain SST-302III-ST, [M-H]^−^ = 479.264, Steviomono-side/Steviol-19-O-glucoside; [M-H]^−^ = 641.316, rubusoside/Steviolbiside; [M-H]^−^ = 803.369, Stevioside. (**B**) RT-PCR analyses of the introduced heterogeneous genes. Additionally, 1, 3, 5, 7, 9, 11, 13, and 15 of the gel lanes from left to right in the figure, were used engineer strains of SST-302III-ST as the cDNA templates to amplification heterologous genes SrCPS, SrKS, SrKO, SrKAH, SrCPR, SrUGT85C2, SrUGT91D2, and SrUGT74G1; 2, 4, 6, 8, 10, 12, 14, and 16 of the gel lanes were using the cDNA template of the original strain BY4742, and the amplification results were used as negative controls. (**C**) Schematic diagram of the glycosylation of stevia.

**Figure 5 microorganisms-12-01125-f005:**
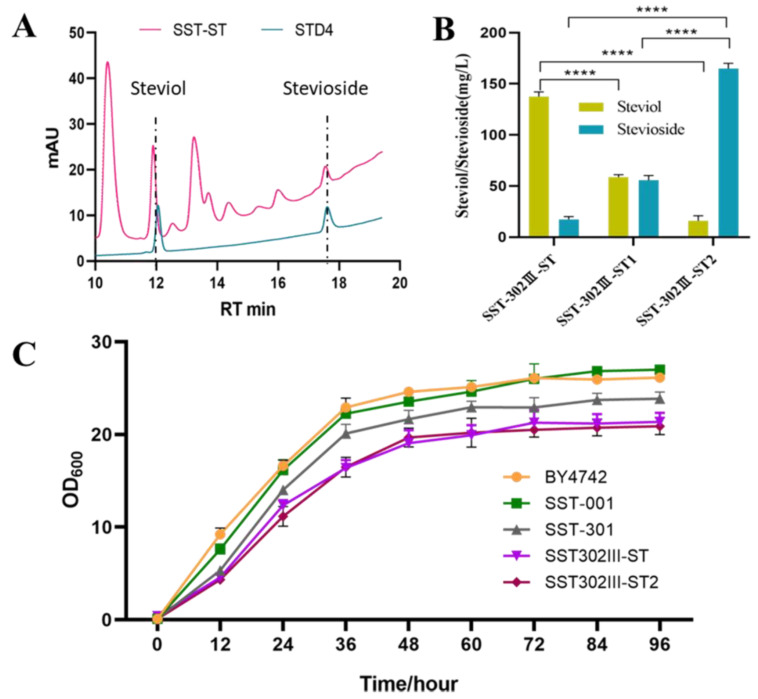
The production of steviside and the biomass of engineering strains. (**A**) Chromatogram of SST-302III-ST and SST-302III-ST1 products. (**B**) Production of stevioside and unconverted steviol in strains SST-302III-ST, SST-302III-ST1 and SST-302III-ST2 (*p* value < 0.0001 is labeled as ****). (**C**) The growth curve of engineered strains.

**Figure 6 microorganisms-12-01125-f006:**
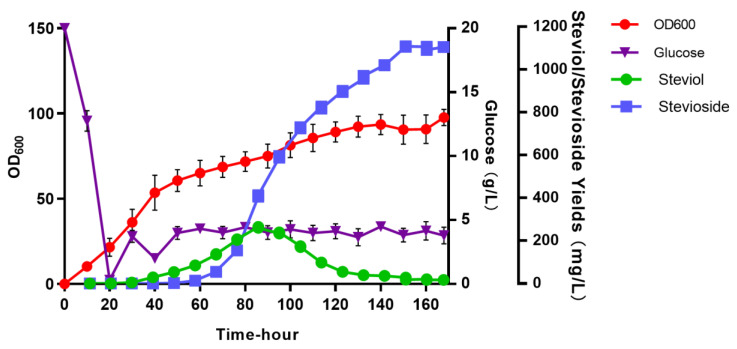
Fed-batch fermentation of yeast strain SST-302III-ST2. The red line represents the OD_600_ of the engineered strain; the purple line represents glucose consumption during fermentation; the green line represents steviol accumulation during fermentation; the blue line represents stevioside accumulation during fermentation.

## Data Availability

The data presented in this study are available on request from the corresponding author.

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
