# Peer review of "Rational Design for the Complete Synthesis of Stevioside in Saccharomyces cerevisiae"

_microorganisms, 2024, doi:10.3390/microorganisms12061125_

Round 1

Reviewer 1 Report

Comments and Suggestions for Authors

I have gone through the manuscript entitled “Rational design for the complete synthesis of stevioside in Saccharomyces cerevisiae”. In this study, authors constructed a Saccharomyces cerevisiae strain for the complete synthesis of stevioside using a metabolic engineering strategy. Despite the problems encountered in the production of the sweeteners, authors carried out a through experimental work and the results are properly presented and discussed. Thus, I recommend publishing the manuscript after minor revision.

1. Title 2.5 “Fed-batch fermentation of yeast strain SST-302III-ST2 was conducted in a 10 L bioreactor” should be shortened to “Fed-batch fermentation of yeast strain SST-302III-ST2”

2. In “3.2. Studies on diterpenoid synthases” authors claim that -(-)ent-kaurene synthases contain an α domain with a DXXDD motif. However, attending to the MSA this motif is better represented by a DDXXD pattern. 

3. What does “AgAS” stand for? Do the authors mean “AgKS”? This is not clear in the manuscript and should be corrected or more clearly expressed to avoid confusion.

4. The provenance of the enzymes should be mentioned as early as possible and not throughout the manuscript. 

Comments on the Quality of English Language

The overall quality of the English language in the manuscript is acceptable; however, minor editing is recommended to enhance clarity and coherence. Some sentences could benefit from a smoother flow and better organization of ideas.

Author Response

Dear Editors and Reviewers,

Thanks for taking your time to review our manuscript entitled “Rational design for the complete synthesis of stevioside in Saccharomyces cerevisiae” (Manuscript ID Microorganisms-3037919). We really appreciate all your careful comments and suggestions! Those comments are all valuable and very helpful for revising and improving our paper, as well as the important guiding significance to our researches. We have studied comments carefully and have made correction which we hope meet with approval.

Response to reviewer #1:

Q1. Title 2.5 “Fed-batch fermentation of yeast strain SST-302III-ST2 was conducted in a 10 L bioreactor” should be shortened to “Fed-batch fermentation of yeast strain SST-302III-ST2”

Response: According to your suggestions, we have corrected the title. The specific content and it has been highlighted in yellow.

Q2. In “3.2. Studies on diterpenoid synthases” authors claim that -(-)ent-kaurene synthases contain an α domain with a DXXDD motif. However, attending to the MSA this motif is better represented by a DDXXD pattern.

Response: Thank you for your careful review. I'm so so sorry, it was my mistake. I've corrected it, and it has been highlighted in yellow.

Q3. What does “AgAS” stand for? Do the authors mean “AgKS”? This is not clear in the manuscript and should be corrected or more clearly expressed to avoid confusion

Response: Thank you for the careful review. The “AgAS” is abietadiene synthase from Abies grandis, and it mainly catalyzes GGPP formed nor-CPP, and then cyclizes it to form a rosin-type diterpene skeleton. The A in “AS” represents abietadiene, and the K in “KS” represents kaurene.

Q4. The provenance of the enzymes should be mentioned as early as possible and not throughout the manuscript.

Response: Thank you for your advice. I have added the provenance of the enzymes in “Materials and Methods”, 2.2 DNA manipulation, and it has been highlighted in yellow.

Comments on the Quality of English Language

The overall quality of the English language in the manuscript is acceptable; however, minor editing is recommended to enhance clarity and coherence. Some sentences could benefit from a smoother flow and better organization of ideas.

Response: Thanks for your advice of english language. We have further revised the english expression, and the revised part is highlighted in green.

Reviewer 2 Report

Comments and Suggestions for Authors

The work is very interesting and I rate it very highly. For me, analyzing the effects at individual stages of research and adapting the strategy of further research depending on the results obtained is of high value.
However, I would like to point out that the culture conditions in the bioreactor may also significantly affect the metabolic pathway and, therefore, the obtained effects and yields.
The authors keep the pH constant. Has the culture been tested under natural pH changes resulting from metabolism? Monitoring pH changes can also provide additional information about metabolism.
The second important factor is the oxygenation level of the substrate. It is worth monitoring the oxygenation level and possibly increasing its intensity, because the lack of sufficient oxygen in the substrate may limit the growth of biomass and affect the level of metabolites.
Additionally, mixing intensity is also an important parameter that may affect the breeding results. On the one hand, it may be a factor influencing the oxygenation level of the culture medium, on the other hand, it may be a source of shear stress (a stress factor for microorganisms).
The above suggestions do not affect the high rating of the article, but are an encouragement to monitor the level of oxygenation or change the strategy of maintaining the pH value during breeding.

detailed comments

Figure 6. Curve description not visible (chart legend)
line 211 - please enter agitation speed

Author Response

Dear Editors and Reviewers,

Thanks for taking your time to review our manuscript entitled “Rational design for the complete synthesis of stevioside in Saccharomyces cerevisiae” (Manuscript ID Microorganisms-3037919). We really appreciate all your careful comments and suggestions! Those comments are all valuable and very helpful for revising and improving our paper, as well as the important guiding significance to our researches. We have studied comments carefully and have made correction which we hope meet with approval.

Response to reviewer #2:

Thank you very much for your review and your valuable comments.

You mentioned the culture conditions in the bioreactor may also significantly affect the metabolic pathway and, therefore, the obtained effects and yields.

The authors keep the pH constant. Has the culture been tested under natural pH changes resulting from metabolism? Monitoring pH changes can also provide additional information about metabolism.

The second important factor is the oxygenation level of the substrate. It is worth monitoring the oxygenation level and possibly increasing its intensity, because the lack of sufficient oxygen in the substrate may limit the growth of biomass and affect the level of metabolites.

Additionally, mixing intensity is also an important parameter that may affect the breeding results. On the one hand, it may be a factor influencing the oxygenation level of the culture medium, on the other hand, it may be a source of shear stress (a stress factor for microorganisms).

Response: Thanks for your comments, the questions are very meaningful to my research, and it is really important to the fermentation yield. We have read a lot of articles about heterogenous synthesis of terpenoids in Saccharomyces cerevisiae. Referring to their parameter ranges, and combined with the bioreactor in our laboratory, and determined the parameters such as pH, dissolved oxygen and stirring speed (specific parameters in 2.5 have been highlighted in yellow).

Wenping Xie constructed Saccharomyces cerevisiae to heterologous product betacarotenoid (Sequential control of biosynthetic pathways for balanced utilization of metabolic intermediates in Saccharomyces cerevisiae). Fermentation was carried out in 5 L bioreactor , and at 30℃ with an agitation speed of 200 to 500 rpm and an airflow rate of 1 vvm to 2 vvm. pH was controlled at 5.0 by automatic addition of 5 M ammonia hydroxide.

Chuanbo Zhang constructed metabolically engineered yeast cell factory to de novo product zerumbone (Production of sesquiterpenoid zerumbone from metabolic engineered Saccharomyces cerevisiae). Fermentation was carried out in 5 L bioreactor, the ventilation rate is 2vvm, and the stirring speed is adjusted by correlating the stirring speed and dissolved oxygen(DO),the agitation speed was maintained at 300-500 rpm, and DO is controlled above 35% by stirring speed. The pH maintained at 5.5, by automatic addition of 2.5 M H2SO4 and 5 M NH4OH.

Q1. Figure6. Curve description not visible (chart legend).

Response: We aologize for our neglect, and we have added the curve description, and it has been highlighted in yellow.

Q2. Line 211 - please enter agitation speed.

Response: Thank you for the suggestion. The ventilation rate is 2vvm, the agitation speed was maintained at 200-500 rpm,and it was adjusted by correlating dissolved oxygen. I have added this part, and it has been highlighted in yellow.